# Thoracic Vertebral Length-to-Height Ratio, a Promising Parameter to Predict the Vertebral Heart Score in Normal Welsh Corgi Pembroke Dogs

**DOI:** 10.3390/vetsci10020168

**Published:** 2023-02-20

**Authors:** Theethad Tangpakornsak, Phasamon Saisawart, Somchin Sutthigran, Kotchapol Jaturunratsamee, Kittipong Tachampa, Chutimon Thanaboonnipat, Nan Choisunirachon

**Affiliations:** 1Department of Surgery, Faculty of Veterinary Science, Chulalongkorn University, Pathumwan, Bangkok 10330, Thailand; 2Department of Physiology, Faculty of Veterinary Science, Chulalongkorn University, Pathumwan, Bangkok 10330, Thailand

**Keywords:** Corgi, healthy, heart, radiograph, vertebral heart score

## Abstract

**Simple Summary:**

The vertebral heart score (VHS) is the most frequent technique for evaluating the canine cardiac size on thoracic radiographs. However, VHS values are varied among dog breeds, possibly due to the characteristics of the thoracic vertebrae. The characteristics of the thoracic vertebrae and the range of the VHS value of the healthy Welsh Corgi Pembroke (Corgis) have never been reported. Therefore, the current study aimed to explore the characteristics of the thoracic vertebrae of the Corgi in comparison with other small-to-medium breed dogs. Among the dog breeds, the characteristics observed through the ratio between the vertebral length and height were different, and the rank of these ratios among the breeds was correlated to the rank of the previously published VHS values. Comparing among the dog breeds, the characteristics of the Corgi’s thoracic vertebrae were more elongated in shape, and the Corgi-specific VHS was significantly lower than the original reported value. Therefore, the characteristics of the thoracic vertebrae might be one of the main factors affecting the normal VHS values among different dog breeds.

**Abstract:**

The vertebral heart score (VHS) is the sum of the ratio of the cardiac dimensions to the number of thoracic vertebrae, starting from the fourth thoracic vertebra (T4) to the intervertebral disk space (IVS). Breed-specific VHSs, in most cases, were different from the original reference value. Characteristics of the thoracic vertebrae and IVS may influence this variation. This study was conducted to investigate the characteristics of the T4 and IVS on the thoracic radiographs of Corgis in comparison with other small-to-medium breed dogs to evaluate the Corgi-specific VHSs in healthy dogs. The ratio of the T4’s length/height (T4L/H) was significantly different among dog breeds but not the IVS between the T4 and T5. The T4L/H was highest in the Shih Tzu and lowest in Beagle dogs. The Corgi-specific VHS obtained from the ventrodorsal radiograph was significantly higher than that from the dorsoventral radiograph, but a significant difference was not observed between the right and left lateral radiographs. In contrast, the Corgi-specific VHS derived from the right lateral thoracic radiograph was significantly lower than the reference value. This may be correlated with the characteristics of the thoracic vertebrae of Corgis, which were slightly higher than those of the other breeds.

## 1. Introduction

The common diagnostic procedures for cardiac diseases include the audible intensity level of a heart murmur, thoracic radiography [1], electrocardiography, and echocardiography [2]. Despite echocardiography being the gold standard for evaluating cardiac dimensions [1,2,3], survey thoracic radiographs are the first-line clinical diagnostic tool for canine patients with suspected cardiac enlargement due to cardiac disease and is a useful method for detecting congestive heart failure in dogs [4,5] because of its ability to provide information on the pulmonary parenchyma and pulmonary vessels (i.e., cardiogenic pulmonary edema and pulmonary vein distension).

To evaluate the cardiac size on thoracic radiography, objective radiographic methods are recommended, particularly for inexperienced veterinarians. Several objective methods, such as the cardiothoracic ratio (CTR), cardiac size-to-intercostal space, and vertebral heart score (VHS), have been proposed. In addition, the radiographic left atrial dimension (RLAD) [6,7,8] and vertebral left atrial size (VLAS) [7,8,9,10,11] were recently proposed to evaluate the left atrial size on the thoracic radiograph. The CTR was adapted from human medicine [12] for use in dogs. This method was reported to be inconvenient for veterinary clinical practice [13,14], particularly in cases affected by cardiovascular and pulmonary problems [14,15]. Additionally, the cardiac size to intercostal space [16,17,18] was also reported to be used with limitations because of several confounding factors, such as the respiratory phase, interbreed variation with different body conformations, and the superimposition of the ribs [18]. Generally, VHSs, which were first introduced in 1995 [14], are commonly applied in clinical practice. Many VHS-related studies have been conducted, including those focused on breed-specific VHSs [19,20,21], those examining VHSs for specific conditions, such as pericardial effusion [22] and canine heartworm disease [23], and those evaluating VHS-related factors, such as sex [24], radiographic position [25], thoracic conformation [26], and body weight (BW) [26], where some results were different from the original study. Moreover, breed-specific VHSs have been continuously investigated over the last two decades, with results reported for up to more than 28 breeds. Most breed-specific VHSs were higher than the original VHS, which is 9.7 ± 0.5 vertebral unit (v), calculated from 100 dogs with non-specific breeds [14]. The dog breeds that were reported to have higher VHSs than the original value were the Beagle [27], Boxer [24], Cavalier King Charles Spaniel [24], English Bulldog [28], Indian Spitz [29], Labrador Retriever [24,29], Pomeranian [28], Pug [28], and Racing Whippet [30].

Recently, the Corgi has been one of the common companion dogs in several countries. Corgis have been reported to experience cardiovascular diseases, both congenital, such as patent ductus arteriosus [31], and acquired heart diseases. Furthermore, Corgis are classified as a chondrodystrophic breed, similar to Miniature Schnauzers and Dachshunds, whose vertebral conformation may have variations. There is less information on the specific radiographic and cardiac parameters of Corgis, particularly the Corgi-specific VHS. Therefore, this study was designed to characterize the fourth thoracic vertebrae of Corgis compared with other small-to-medium popular chondrodystrophic dog breeds and to explore the VHS range in healthy Corgis. The hypothesis of this study was that Corgi-specific VHSs might be similar to those previously reported on the Dachshund’s VHS because of their similar body conformation and similar thoracic vertebral characteristics.

## 2. Materials and Methods

### 2.1. Part I: A Retrospective Study of Thoracic Radiographs to Compare the Vertebral Characteristics of Corgis with Those of Other Small-to-Medium Dog Breeds

This part was a retrospective study to investigate the characteristics of the fourth thoracic vertebrae of healthy Corgis on the right lateral (RL) thoracic radiograph and compare them with those from four other small-to-medium dog breeds, such as Beagles, Dachshunds, Schnauzers, and Shih Tzus. Lateral thoracic radiographs from 20 dogs from each aforementioned breed taken between January 2018 and December 2020 in the Digital Imaging and Communications in Medicine (DICOM) format were retrieved from the Diagnostic Imaging Unit, the Small Animal Hospital, Faculty of Veterinary Science, Chulalongkorn University. All data in this section were approved by the hospital board committee, the Small Animal Hospital, Faculty of Veterinary Science, Chulalongkorn University, with approval no. S152/2564. The inclusion criteria included lateral thoracic radiographs from dogs of each breed whose age was between 1 and 8 years without a history and radiographic signs of vertebral and intervertebral diseases. In any case of the thoracic radiographs presenting vertebral abnormalities, intervertebral diseases, or an improper radiographic position and technique, the dog was excluded from this study. In addition, Corgis with a history of cardiac diseases based on clinical signs, radiographic evidence of cardiomegaly or pulmonary edema, and echocardiographic reports were excluded from the study. The length and height of the T4 (T4L and T4H) and the IVS between the T4 and T5 (T4-T5 IVS) were measured on lateral radiographs (Figure 1) using a digital caliper on DICOM viewer version 5.6 software (OSIRIX^®^, Geneva, Switzerland) by an experienced radiologist with more than 10 years of experience (TT). The T4L and T4H were measured at the central area of the vertebral body of the T4, which was parallel and perpendicular to the long axis, respectively. The T4 length-to-height ratio (T4L/H) was then calculated. Because of the different body sizes of each dog breed, only the T4L/H and T4-T5 IVS were compared among the dog breeds.

### 2.2. Part II: A Prospective Study to Investigate Corgi-Specific VHSs

The second part was designed as an observational prospective study. The study was approved by the Institutional Animal Care and Use Committee of Chulalongkorn University (no. 2131020), and written consent was received from the dogs’ owners. Healthy Corgis that visited the Diagnostic Imaging Unit, Small Animal Hospital, Faculty of Veterinary Science, Chulalongkorn University, between August 2021 and February 2022 were recruited for this study. The inclusion criteria were healthy Corgis, both male and female, aged between 2 and 5 years, and with a body condition score (BCS) of 4–6/9. Dogs with heart and/or other cardiovascular diseases, lung abnormalities, and immature or thoracic vertebral abnormalities were excluded from this study. Subsequently, all signalments and histories of the included dogs were then obtained. Then, all dogs underwent a full physical examination, including appearance, body temperature, dehydration status, mucus membrane color, heart and respiratory rates, heart and lung sounds, blood pressure, and whole-body palpations. Subsequently, a complete blood count, biochemistry panel, and thoracic radiographic and echocardiographic examinations were performed. If any dogs revealed abnormal results based on physical or blood examinations, they were excluded from this study.

### 2.3. Radiography and Screening Echocardiography

Four positions of thoracic radiographs, including the RL, the left lateral (LL), the ventrodorsal (VD), and dorsoventral (DV) views of all dogs were taken without sedation by an experienced cardiologist with more than 20 years of experience (KT). All dogs were positioned carefully during the radiography in an attempt to obtain a radiograph at a full inspiratory phase. All radiographs were obtained using a digital radiography system (Brivo DR-F, GE Healthcare, Chicago, IL, USA) and saved in the DICOM format. Furthermore, an echocardiographic examination was performed on all enrolled dogs without sedation to confirm normal cardiac morphology and function. An M9 Mindray^®^ ultrasound machine (Shenzhen, China) was used to record echocardiograms using a phased-array ultrasound probe (Mindray^®^ P7-3s) for all right and left parasternal axes. If any dogs showed any abnormal radiographic or echocardiographic signs, they were excluded from this study.

### 2.4. VHS Measurement

The VHS was measured using a digital caliper in a digital format, as previously reported [14] in all radiographic views using the similar DICOM viewer software as in part I by an experienced radiologist with more than 10 years of experience (TT). In brief, on a lateral radiograph, a long axis of the cardiac silhouette was drawn from the ventral border of the left main stem bronchus to the cardiac apex, and a short axis was drawn from a line perpendicular to the long axis at the same level of the mid diameter of the caudal vena cava [14]. The long and short axes on the VD or DV radiograph were drawn from the maximum length of the heart and the perpendicular widest dimension. Then, both lines from each radiographic view were transposed over the thoracic spine, starting at the cranial aspect of the T4 (parallel to the vertebral canal) on the RL view. The summation of these lines was translated into vertebral units to the nearest 0.1 vertebrae (v). Subsequently, VHSs were compared depending on each factor, such as radiographic positionings, sex, and BW.

### 2.5. Statistical Analysis

All data in this study were analyzed by Prism7 software (GraphPad, La, Jolla, CA, USA). The Shapiro–Wilk and D’Agostino–Pearson omnibus normality tests were used to evaluate the normality of the data of the first and second parts of the study, respectively, and the results were expressed as means ± standard deviations and 95% prediction intervals if the dataset had a normal distribution. An analysis of variance and post hoc Tukey test were used to compare the ratios of the T4L/H and T4-T5 IVSs among the breeds. A one-sample *t*-test was used to test the difference between the VHS of Corgis and the mean reference values proposed by Buchanan and Bücheler [14]. An unpaired *t*-test was used to compare the T4L, the T4H, the T4-T5 IVS between the sexes in each breed, and the T4L/H between the sexes, including the VHS between the sexes and the radiographic positions, RL, LL, VD, and DV, respectively. A *p* < 0.05 was considered statistically significant.

## 3. Results

### 3.1. Part I: A Retrospective Study of Thoracic Radiographs to Compare the Vertebral Characteristics of Corgis with Those of Other Small-to-Medium Dog Breeds

The clinical demographic data of Beagles, Corgis, Dachshunds, Schnauzers, and Shih Tzus in part I are shown in Table 1. All the data in this part were normally distributed, and the mean age of all the dogs was 7.37 ± 3.38 years. Significant differences in age were observed among the breeds (*p* = 0.0025). The mean age of the Shih Tzus was the highest, whereas that of the Beagles was the lowest. Additionally, the mean BW among the dog breeds was 10.87 ± 4.62 kg. Moreover, significant differences in BW were observed among the breeds (*p* < 0.0001). The mean BW of the Beagles was the highest, whereas that of the Shih Tzus was the lowest. Furthermore, no statistical differences in age and BW were observed between the sexes (*p* = 0.36 and *p* = 0.17, respectively).

On the RL radiographs, the mean T4L, T4H, and T4L/H of the Beagles, Corgis, Dachshunds, Schnauzers, and Shih Tzus are shown in Table 2. The mean T4L and T4H of all the dogs were 1.32 ± 0.17 and 0.88 ± 0.14, respectively. The mean of the T4L of the Corgis and Schnauzers was the highest and lowest, respectively. The T4L values between the sexes of all the dog breeds were not significantly different. In contrast, the Beagles and Shih Tzus had the highest and lowest mean T4H values, respectively. The T4H between the sexes of almost all the dog breeds was not significantly different, except for female Dachshunds, which had a higher T4H than the male dogs (*p* = 0.0305). Additionally, the T4L/H ratio was significantly different among the dog breeds. The highest T4L/H value was found in the Shih Tzus, followed by those of the Corgis, Dachshunds, Schnauzers, and Beagles, respectively (*p* < 0.0001, Figure 2). Moreover, the T4L/H ratio between the males and females in each dog breed was not significantly different (Shih Tzu, *p* = 0.12; Corgi, *p* = 0.08; Schnauzer, *p* = 0.49; Beagle, *p* = 0.43; and Dachshund, *p* = 0.24).

Additionally, the mean lengths of the T4-T5 IVS on the lateral radiographs of the Corgis, Shih Tzus, Schnauzers, Beagles, and Dachshunds are also shown in Table 2. The mean T4-T5 IVS of all the dogs was 0.16 ± 0.04; the mean T4-T5 IVS of the Dachshunds was the highest, whereas that of the Schnauzers was the lowest. However, the length of the T4-T5 IVS among the Corgis, Shih Tzus, Schnauzers, Beagles, and Dachshunds was not significantly different (*p* = 0.13). Furthermore, no statistical difference in the T4-T5 IVS was observed between the sexes, except for male Corgis, which had a wider IVS than female Corgis (*p* = 0.03).

### 3.2. Part II: A Prospective Study to Investigate the Corgi-Specific VHS

From September 2021 to December 2021, 31 Corgi dogs were enrolled in this study. However, one dog was excluded because of radiographic evidence of thoracic hemivertebrae. Therefore, 30 healthy Corgis were included and divided into sex groups (15 male and 15 female dogs). All the data in this part were normally distributed, and the mean age and BW were 34.13 ± 17.68 months and 12.97 ± 2.99 kg, respectively. The mean BCS was 4.90 ± 1.12. In this study, the BW of the Corgi dogs was not significantly different between the sexes (*p* = 0.16).

### 3.3. VHS on the Thoracic Radiograph

In this study, the mean Corgi-specific VHSs, including those on the RL, LL, VD, and DV radiographs, are presented in Table 3. No significant difference in the VHSs was observed between the RL and LL positions (*p* = 0.1913). Conversely, the VHSs on the VD radiographs were significantly higher than those on the DV radiographs (*p* = 0.0212, Figure 3). Furthermore, the VHSs between the sexes in all the radiographic positions were not significantly different (RL, *p* = 0.84; LL, *p* = 0.92; VD, *p* = 0.43; and DV, *p* = 0.96).

The linear regression analysis showed no correlation between the VHS and BW in all of the radiographic positions (RL, *r* = 0.27, *p* = 0.15; LL, *r* = 0.07, *p* = 0.73; VD, *r* = 0.13, *p* = 0.48; and the DV position, *r* = 0.23, *p* = 0.20). Additionally, the breed-specific VHS from adult Corgis observed on the RL thoracic radiographs was 9.36 ± 0.27 v., which was significantly lower than the VHS from that reported in the original reference studies by Buchanan and Bücheler (9.7 ± 0.5 v.) [14] (*p* < 0.0001). In contrast, the VHSs measured from the VD and DV radiographs (10.02 ± 0.57 v. and 9.84 ± 0.64 v., respectively) were not statistically different from the original values (10.02 ± 0.83 v. and 10.02 ± 1.45 v., respectively) (*p* = 1 and *p* = 0.13, respectively) [14].

## 4. Discussion

Breed-specific VHSs are recommended to evaluate the radiographic cardiac size [32]. Currently, up to 28 dog breeds have been studied and reported on. However, there was no report related to Corgi-specific VHSs. Therefore, this is the first study to investigate Corgi-specific VHSs, one of the chondrodystrophic dog breeds. Additionally, this study explained the variability in the size and shape of the T4, which may be the main factor impacting the variability in the breed-specific VHS. The variation in breed-specific VHSs may be because they are an objective radiographic method. The VHS is the sum of the ratio between the length and the width of the cardiac silhouette to the number of thoracic vertebrae, starting from the fourth thoracic vertebra (T4) to the intervertebral disk space (IVS). Therefore, the interbreed variations in the thoracic vertebrae caused variations in the VHS values, suspected to be because of the variations in the size and shape of the thoracic vertebrae and IVS values. Several variations in the thoracic vertebrae have been reported, such as naturally short vertebrae in Miniature Schnauzers [14,33] or normal dogs with a long-chested look, such as Dachshunds. These breeds tend to have different VHSs from the original value [14,34].

In this study, the ratio of the T4L and T4H from the lateral radiographs was used for two reasons. First, the VHS was described to compare the radiographic cardiac size with the length of the thoracic vertebrae, starting from the T4, which is commonly observed on lateral radiographs. The actual T4L is suggested not to be compared among dog breeds because of the variability in body size that may directly be correlated with the dimensions of the vertebral column. Furthermore, this study revealed no significant differences in the T4L and T4H between the sexes in each breed, except for Dachshunds, where females had a higher T4H than males. This discrepancy indicated that Dachshunds have one of the dimorphisms bred between the sexes. However, further studies to investigate the vertebral characteristics between the sexes that may affect the VHS should be conducted in a larger population. Second, the actual size of the T4 on the VD view was difficult to measure compared with those on the lateral radiographs because of the poor demarcation of the vertebrae and superimposition with other skeletal structures, such as the sternum.

The T4L/H ratio was then selected to be compared and observed for breed variations. The higher T4L/H ratios indicated that the vertebrae tend to be more rectangular in shape. In contrast, the vertebrae tend to be square in shape when the T4L/H ratio is low. Comparing the five breeds, Shih Tzus had the highest, followed by Corgis, Dachshunds, and Schnauzers, whereas the lowest T4L/H ratios were detected in Beagles. This T4L/H indicated that Shih Tzus tend to have the longest thoracic vertebrae, whereas Beagles have the shortest T4. Schnauzers were reported to be a naturally short-vertebrated dog breed [33], and the normal Schnauzer-specific VHS reached 11 [14]. Nevertheless, the exact value of Schnauzer-specific VHSs has not yet been published. Furthermore, this finding was correlated with previous breed-specific VHSs, reporting that normal Shih Tzus had a lower VHS (9.30 ± 0.50 v.) [25] and normal Beagles had a higher VHS (10.50 ± 0.40 v.) [27] than that (9.70 ± 0.50 v.) reported in the original study [14]. Additionally, the breed-specific VHS of the Dachshunds was 9.70 ± 0.50 v. [28], which is similar to the reference values reported in the original study. In this study, the T4L/H ratio of the Dachshund was 1.46 ± 0.16, which was in the middle value among these breeds. Therefore, the T4L/H ratio may be useful as a predictor in cases where breed-specific VHSs are unavailable. In this study, the VHS tended to be lower than 9.70 ± 0.50 if the T4L/H ratio was higher than 1.46, and the VHS tended to be higher than 9.70 ± 0.50 v. if the T4L/H ratio was lower than 1.46. Furthermore, the T4L/H comparison between the sexes revealed no significant difference in these five dog breeds, indicating that this parameter is useful for use without the effect of sexual dimorphism. However, studies on this issue in a larger population with various dog breeds are required.

In addition to the VHS being impacted by the characteristics of the T4, novel radiographic objective methods, such as RLAD [6] and VLAS [9], also use the thoracic vertebral length for the evaluation process. These methods were used to evaluate the heart size, focusing on the left atrial dimension, which is commonly enlarged secondary to degenerative mitral valve disease. It has been reported that Cavalier King Charles Spaniels [35], Chihuahuas [19], Maltese [11], and Pug-specific VLAS [7] have a lower value than the original value reported by Malcolm [9]. Therefore, the LA assessment using these two techniques should be warranted if there is a variation in the thoracic vertebral characteristics indicated by the T4L/H ratio. In addition to the T4L/H ratio, the length of the IVS is a possible factor that may affect the VHS. In this study, the IVS is varied among the dog breeds, possibly due to a wide range of body sizes. Therefore, further investigation of the IVS in a large population of dogs with different body sizes is also suggested. Nevertheless, this study revealed no significant difference in the IVS among these breeds and between the sexes in each breed, except for Corgis, in which male Corgis had a wider IVS than female Corgis. According to the American Kennel Club, female Corgis can weigh up to 28 pounds, whereas males may reach 30 pounds [36], indicating that there is a sexual dimorphism of the body phenotype between the sexes.

The VHS of Corgis in this study was 9.36 ± 0.27 v., which is also lower than 9.70 ± 0.50 v. Although the body conformation of Corgis is similar to that of Dachshunds, the VHSs were different. This evidence may correlate with the T4L/H ratio, where Corgis tended to have a higher T4L/H value than Dachshunds and other breeds. In this study, no statistically significant difference in Corgi-specific VHSs was observed between the RL and LL radiographs. This finding was similar to that reported in previous studies using several dog breeds, except for Whippets [30], Shih Tzus [25], Beagles [27], the Indian Spitz, Labrador Retrievers [29], Dachshunds [20], and Australian Cattle dogs [21], where the VHSs retrieved from the RL radiographs were higher than those on the LL radiographs. Conversely, the Corgi-specific VHSs from the VD radiographs were significantly higher than those obtained from the DV position, which was similar to those in the original study [14], Whippets [30], and Australian Cattle dogs [21]. The dimensions of the cardiac silhouette in the VD radiograph were increased by 7.2% and 5.3% in width and length, respectively, compared with those on the DV view [14]. The most reliable explanation for this result is the magnification associated with a greater cardiac distance in relation to the cassette or digital detector on the VD view. However, a lateral radiograph was preferred for several reasons, such as the feasibility of the radiographic positioning, particularly in patients with dyspnea [37]. Furthermore, several dog breeds were reported to have dimorphism effects, such as Boxers, Labrador Retrievers, Cavalier King Charles Spaniels, Dobermans, German Shepherds [24], Yorkshire Terriers [24,28], and Dachshunds [20]. No significant difference between the VHS and the sexes of Corgis was found in this study. This indicated that Corgis had no dimorphism effect on the VHS between the sexes, similar to several dog breeds.

There are some limitations to this study. The overall limitation for both study parts was the small population size. In addition, all of the dogs were evaluated only for T4 characteristics and were not evaluated for the self-VHS in the first part because a retrospective model could not be used by the authors to confirm the normal cardiac morphology and cardiac function of all the dogs through the echocardiographic examination. Therefore, further VHS studies, including the T4L/H, among various dog breeds should be prospectively conducted. Moreover, the T4L could also not be compared due to the different body sizes. In part II, in addition to the limitation of the small population size, the correlation of the Corgi-specific VHSs among the age groups and between healthy and diseased groups was not evaluated.

## 5. Conclusions

In conclusion, thoracic vertebral characteristics are various among dog breeds and may affect the VHS, including the other radiographic measurements that compare the cardiac dimensions to the thoracic vertebrae. Corgis have a higher T4L/H ratio, which means that the thoracic vertebrae are slightly longer, and revealed a significantly lower VHS (9.36 ± 0.27 v.) than the previous original reference range (9.70 ± 0.50 v.).

## Figures and Tables

**Figure 1 vetsci-10-00168-f001:**
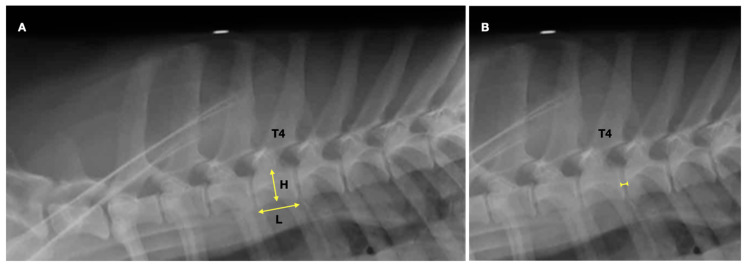
The measurement method of the fourth thoracic vertebral (T4) length and height (**A**) and the length of the fourth and the fifth intervertebral disc space (**B**) on the lateral thoracic radiograph.

**Figure 2 vetsci-10-00168-f002:**
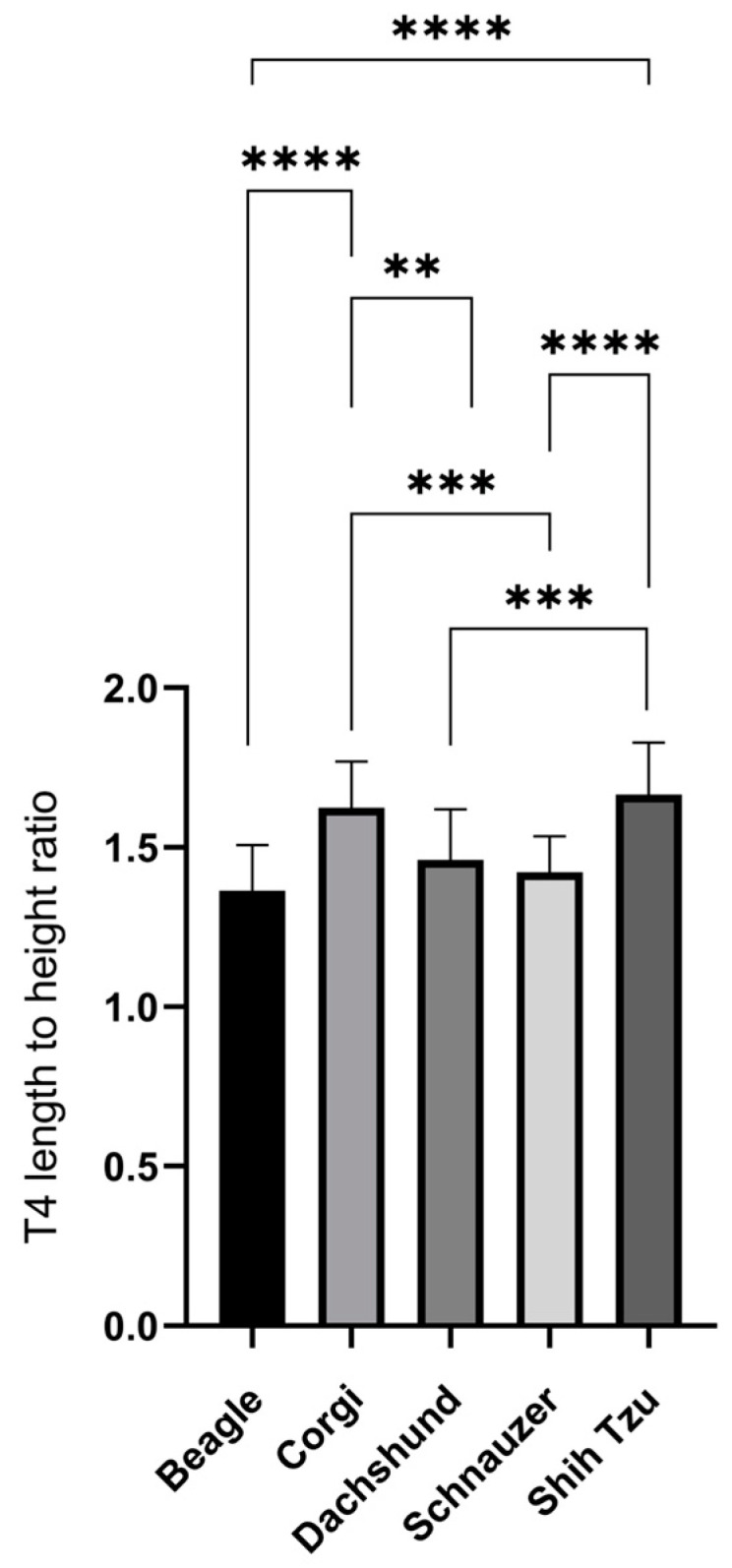
The length-to-height ratio of the fourth thoracic vertebrae among Corgis, Shih Tzus, Schnauzers, Beagles, and Dachshunds. Data were expressed as mean ± SEM. ** *p* < 0.01; *** *p* < 0.001; **** *p* < 0.0001.

**Figure 3 vetsci-10-00168-f003:**
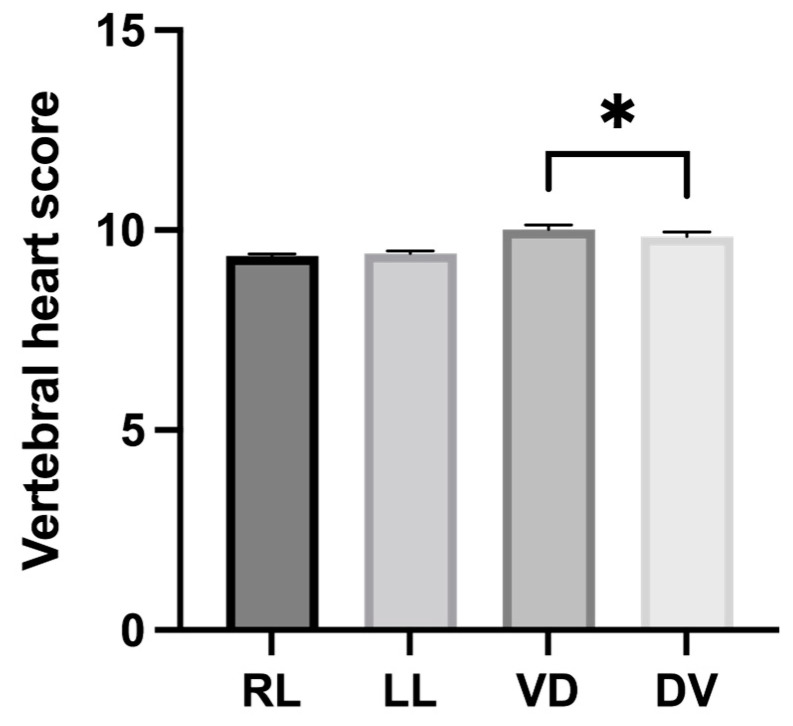
The mean difference of Corgi-specific vertebral heart scores between right lateral (RL), left lateral (LL), ventrodorsal (VD), and dorsoventral (DV) positions (* significantly different, *p* < 0.05).

**Table 1 vetsci-10-00168-t001:** Clinical demographic information, including sex, age, and body weight (BW) of all dog breeds. Data were expressed as mean ± SD.

	Beagle	Corgi	Dachshund	Schnauzer	Shih Tzu
Number	20	20	20	20	20
Sex					
Male	12	9	7	9	8
Female	8	11	13	11	12
Age (years)	5.65 ± 3.05 ^a^ ** ^b^ *	6.60 ± 3.80	6.75 ± 3.65	8.65 ± 3.08 ^b^ *	9.20 ± 1.94 ^a^ **
BW (kg)	15.45 ± 2.96 ^d, f, g^ ***	14.09 ± 4.72 ^c, e^ *** ^i^ **	10.47 ± 3.62 ^g^ *** ^h, i^ ** ^j^ *	7.43 ± 1.67 ^e, f^ *** ^j^ *	6.89 ± 1.59 ^c, d^ *** ^h^ **

* Significant difference at *p* value < 0.05; ** significant difference at *p* value < 0.01; *** significant difference at *p* value < 0.001; ^a^ significant difference of age between Beagle and Shih Tzu. ^b^ Significant difference of age between Beagle and Schnauzer. ^c^ Significant difference of BW between Corgi and Shih Tzu. ^d^ Significant difference of BW between Beagle and Shih Tzu. ^e^ Significant difference of BW between Corgi and Schnauzer. ^f^ Significant difference of BW between Beagle and Schnauzer. ^g^ Significant difference of BW between Beagle and Dachshund. ^h^ Significant difference of BW between Dachshund and Shih Tzu. ^i^ Significant difference of BW between Corgi and Dachshund. ^j^ Significant difference of BW between Dachshund and Schnauze.

**Table 2 vetsci-10-00168-t002:** The length and the height of the fourth thoracic vertebrae (T4L and T4H; cm) and the T4 length-to-height ratio (T4L/H) and the intervertebral disc space (IVS; cm) on the lateral radiograph among the Corgi, Shih Tzu, Schnauzer, Beagle, and Dachshund.

	Beagle	Corgi	Dachshund	Schnauzer	Shih Tzu
T4L (cm)	1.33 ± 0.12	1.51 ± 0.08	1.39 ± 0.13	1.14 ± 0.07	1.23 ± 0.12
Male	1.35 ± 0.13	1.51 ±0.09	1.44 ± 0.16	1.17 ± 0.08	1.27 ± 0.15
Female	1.31 ± 0.13	1.51 ±0.08	1.36 ± 0.12	1.11 ± 0.05	1.20 ± 0.09
T4H (cm)	0.98 ± 0.11	0.93 ± 0.08	0.96 ± 0.12	0.80 ± 0.08	0.74 ± 0.08
Male	0.99 ± 0.10	0.97 ± 0.08	1.44 ± 0.15 ^a^ *	0.83 ± 0.08	0.75 ± 0.10
Female	0.95 ± 0.11	0.90 ±0.08	1.36 ± 0.12 ^a^ *	0.78 ± 0.08	0.74 ± 0.08
T4L/H	1.36 ± 0.14 ^e^ *^, f^ *	1.62 ± 0.14 ^b^ *^, d^ *^, e^ *	1.46 ± 0.15 ^b^ *^, c^ *	1.42 ± 0.11 ^d^ *^, g^ *	1.66 ± 0.16 ^c^ *^, f^ *^, g^ *
Male	1.35 ± 0.11	1.56 ± 0.14	1.39 ± 0.13	1.41 ± 0.08	1.69 ± 0.07
Female	1.38 ± 0.18	1.67 ± 0.13	1.49 ± 0.16	1.43 ± 0.13	1.64 ± 0.20
IVS (cm)	0.15 ± 0.03	0.16 ± 0.03	0.17 ± 0.05	0.14 ± 0.03	0.16 ± 0.03
Male	0.15 ± 0.02	0.18 ± 0.02 ^h^ *	0.18 ± 0.04	0.14 ± 0.04	0.16 ± 0.03
Female	0.14 ± 0.03	0.14 ± 0.03 ^h^ *	0.16 ± 0.06	0.14 ± 0.03	0.15 ± 0.04

* Significant difference at *p* value < 0.05. ^a^ Significant difference in T4H between the sexes of Dachshunds. ^b^ Significant difference of T4L/H between Corgis and Dachshunds. ^c^ Significant difference of T4L/H between Shih Tzus and Dachshunds. ^d^ Significant difference of T4L/H between Corgis and Schnauzers. ^e^ Significant difference of T4L/H between Corgis and Beagles. ^f^ Significant difference of T4L/H between Shih Tzus and Beagles. ^g^ Significant difference of T4L/H between Shih Tzus and Schnauzers. ^h^ Significant difference of IVS between the sexes of Corgis.

**Table 3 vetsci-10-00168-t003:** Vertebral heart scores of Corgis on all radiographic positions of right lateral (RL), left lateral (LL), ventrodorsal (VD), and dorsoventral (DV) projections.

Position	Mean ± SD	95%CI
RL		
All	9.36 ± 0.27	9.25–9.45
Male	9.36 ± 0.26	9.22–9.51
Female	9.34 ± 0.29	9.18–9.51
LL		
All	9.42 ± 0.37	9.28–9.55
Male	9.42 ± 0.33	9.24–9.61
Female	9.41 ± 0.41	9.18–9.64
VD		
All	10.02 ± 0.57 ^a^	9.81–10.24
Male	9.94 ± 0.54	9.63–10.24
Female	10.11 ± 0.59	9.77–10.44
DV		
All	9.84 ± 0.64 ^a^	9.59–10.08
Male	9.84 ± 0.71	9.45–10.24
Female	9.83 ± 0.58	9.50–10.16

^a^ VD and DV VHSs differed significantly (*p* = 0.0212) in Corgis.

## Data Availability

Raw data that supported the finding of this study are available from the corresponding author upon reasonable request.

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
