# Peer review of "Thoracic Vertebral Length-to-Height Ratio, a Promising Parameter to Predict the Vertebral Heart Score in Normal Welsh Corgi Pembroke Dogs"

_vetsci, 2023, doi:10.3390/vetsci10020168_

Round 1

Reviewer 1 Report

The reviewer would like to to thank the authors for this interesting acticle and appreciate the effort. Overall, the citiations need to be revised. Current studies are missing, especially from the last years. The missing studies compared different radiographic measurement (including VHS). 

Please do not abbreviate the breed in the title and first time use in the article. Was the breed the "Welsh Corgi Pembroke" or the "Welsh Corgi Cardigan"? Please specify. 

Line 11

Add "The" vertebral heart score

Line 12: 

Please change "heart" to "cardiac".

Line 18-19:

The sentence appears incompllete. Please rewrite.

Line 26:

Please indicate whether the ratio was higher or lower.

Line 37:

The sentence about cardiac computed tomographic shloulb be deleted, because it is irrelevant for this study. 

Line 38: 

Please add a citiation.

Line 44:

Please include that a pulmonary edema can be detected with radiography.

Line 45:
Please change heart to cardiac.

Line 46-50:

Please add the radiographic methods VLAS and RLAD, because you mention them in the dicussion. Addionally, the VLAS shoulb be mention, because it is mentioned in the ACVIM Consensus Statement. To shorten the article CTR can be deleted.  

Line 55-59

Please add references

Line 61-62

Pelase give an example for breeds with higher VHS

Line 62

The term vertebrae as unit is commonly used. However , is not accurate, because the intervertebral space (as mentioned in the article) is included. For example the term Vertebral Units can be used or vertebrae can be deleted.

Line 63

The possible reason for variation in breed-specific VHS shoulb be mentioned in the discussion. Please revise.

Line 72

Please delete the term "one of a fashionable dog breed". Is there any evidence that Corgi become more popular?

Line 78-80

Please change (i) and (ii) to first ans second.

Line 89-90

Please list the breed in alphabetical order.

Matereial and Methods Part I:

Did the owners give their permission for using the radiographs of their dogs?

Please confirm that every dog was evaluted with echocardiography. 

Who performed the radiographic measurements?

Material and Methods Part II

Again, who performed the raiographic measurements? And who performed the echocardiographic evaluation?

Line 145

Please specify the level of the caudal vena cava.

Line 150

The explanation of the unit vertebrae is good. Maybe you can use it for line 62.

Statistical analysis

Which software was used. Please specify.

Line 164

Please add "A p-value<0.05" or "A p<0.05".

Results:

Please confirm that all values are normally distibuted

Line 169

Please list the breed in alphabetical order and later on, too.

Table 1 and Figure 2.

Please write "Shih Tzu"

Table 2

Shoulb it be p<0.05? Please confirm.

Table 3

A range is not needed for normally distrubuted values.

Figure 3.

Please add the meaning of the asterik.

Line 249:

Please change heart to cardiac.

Line 291

Using defined measurement poits, RLAD and VLAS are objevtive methods like VHS.

Line 293

Please explain the abbreviation LA. 

Line 295 

VLAS reference values for Pugs, Maltese, and Cavalier King Charles Spaniel hab been published.

Line 302:

Where the males Corgis larger?

Line 312-313:

Mongrels are not define and therefore not suitbale for a comparision . Please use as reference for the Laprador Retriever  Lamb et al. [2001].

Line 323-328

I think this sentence can be deleted, because it provides no necceassy information for the discussion (and LV should be explained).

Limations

As the authors mentioned in the disscusion that study which larger groups should be conducted, the population size shoulb be mention as limitation. 

a second limitaion is that there was no comparision between the breeds for T4L as mention in Line 258.

A measurement example of VHS can be inclueded.

For example following citations can be included:

https://doi.org/10.1371/journal.pone.0274085

https://doi.org/10.1016/j.jvc.2020.06.001

https://doi.org/10.1016/j.jvc.2022.08.004

https://doi.org/10.1111/vru.13027

https://doi.org/10.1136/vr.148.23.707

Author Response

Reviewer1

The reviewer would like to thank the authors for this interesting article and appreciate the effort. Overall, the citations need to be revised. Current studies are missing, especially from the last few years. The missing studies compared different radiographic measurements (including VHS). 

 Response: Thank you very much for all comments and suggestions. Authors have revised and responded to all comments with the following details.

Please do not abbreviate the breed in the title and first time use in the article. Was the breed the "Welsh Corgi Pembroke" or the "Welsh Corgi Cardigan"? Please specify. 

Response: This study was done in Welsh Corgi Pembroke. Therefore, authors have revised the title and at the first time to indicate the dog breed by inserting the full name at line 3-4 and 15. 

Line 11 Add "The" vertebral heart score

Response: Authors added “the” at line 12.

Line 12: Please change "heart" to "cardiac".

Response:  Authors revised from “heart” to “cardiac”  at line 13.

Line 18-19: The sentence appears incomplete. Please rewrite.

Response: The original file the authors submitted to the system was complete. We think that this error might be the result of some correction from the editorial system. However, we rechecked the original file and corrected it in the latest version downloaded from the system, line 20-21. 

Line 26: Please indicate whether the ratio was higher or lower.

Response: The ratio was highest in Shih Tzu and lowest in Beagle. Authors added this information at line 31.

Line 37: The sentence about cardiac computed tomography should be deleted, because it is irrelevant for this study. 

Response: Authors removed this sentence from introduction following reviewer’s suggestion.

Line 38: Please add a citation.

Response: Authors added citation at line 41-42 following reviewer’s suggestion.

Line 44: Please include that a pulmonary edema can be detected with radiography.

Response: Authors added the information of pulmonary edema in the following context “because of its ability to provide information on the pulmonary parenchyma and pulmonary vessels (i.e. cardiogenic pulmonary edema and pulmonary vein distension)” at line 45-47.

Line 45: Please change heart to cardiac.

Response: Authors revised from “heart” to “cardiac”  at line 48.

Line 46-50: Please add the radiographic methods VLAS and RLAD, because you mention them in the discussion. Additionally, the VLAS should be mentioned, because it is mentioned in the ACVIM Consensus Statement. To shorten the article CTR can be deleted.  

Response: Authors added the information of RLAD and VLAS as following context “IIn addition, radiographic left atrial dimension (RLAD) [6-8] and vertebral left atrial size (VLAS) [7-11] were recently proposed to evaluate the left atrial size on the thoracic radiograph.” at line 51-53.

Line 55-59 Please add references

Response: Authors added references at line 62-65.

Line 61-62 Please give an example for breeds with higher VHS

Response: Authors added examples of dog breeds with higher VHS at line 68-71.

Line 62 The term vertebrae as a unit is commonly used. However , it is not accurate, because the intervertebral space (as mentioned in the article) is included. For example the term Vertebral Units can be used or vertebrae can be deleted.

Response: Authors revised the term from “vertebrae” to be “vertebral unit” following reviewer’s suggestion at line 67.

Line 63 The possible reason for variation in breed-specific VHS should be mentioned in the discussion. Please revise.

Response: Authors revised by moving this paragraph and inserted in the discussion at line 289-297.

Line 72 Please delete the term "one of a fashionable dog breed". Is there any evidence that Corgi has become more popular?

Response: In Thailand, Corgi’s are recently becoming a  popular breed. However, to make the context clear and can be understandable worldwide, authors decided to revise the context in this paragraph to be “Corgi is one of the common raising companion dogs in several countries.” at line 72

Line 78-80 Please change (i) and (ii) to first and second.

Response: To make the context easier to understand, authors revised this context by removing (i) and (ii) at line 78 and 79. 

Line 89-90 Please list the breed in alphabetical order.

Response: Authors revised the manuscript following authors suggestion at line 88-89.

Matereial and Methods Part I:

Did the owners give their permission for using the radiographs of their dogs? 

Response: The data in the first part was a retrospective information retrieved from the teaching hospital belonging to the University, the approval no: S152/2564. All data were allowed to be used by the hospital board committee and authors add the information at line 93-95.

Please confirm that every dog was evaluated with echocardiography. 

Response: In the first part, all thoracic radiographs will be observed only for the characterization of the thoracic vertebrae not the cardiac information. Therefore, the inclusion criteria and exclusion criteria involved only the necessary information about thoracic vertebrae and echocardiographic information not retrieved in all dogs.

Who performed the radiographic measurements?

Response: All radiographic measurements were performed by the first author (TT) who is an experienced radiologist with more than 10 years of experience. Authors added this information at line 104-105. 

Material and Methods Part II

Again, who performed the radiographic measurements? And who performed the echocardiographic evaluation?

Response: Radiographic measurement is done by the same author as part I (TT) and echocardiographic evaluation was performed by  KT who is a 20 year-experience cardiologist. Authors added the information of who were performing echocardiographic examination at line 134-135. and the information of who were performing VSH measurement at line 148.

Line 145 Please specify the level of the caudal vena cava.

Response: Authors specify the level of the caudal vena cava as shown in line 151-152. The level of caudal vena cava was as same as the original study.

https://www.vin.com/apputil/content/defaultadv1.aspx?pId=84&id=4253805

Line 150 The explanation of the unit vertebrae is good. Maybe you can use it for line 62.

Response: Thank you very much, authors applied vertebral units at both locations. 

Statistical analysis

Which software was used. Please specify.

Response: Authors added the information about software at line 161-162.

Line 164 Please add "A p-value<0.05" or "A p<0.05".

Response: Authors added “A” in front of the p<0.05 at line 171.

Results:Please confirm that all values are normally distributed

Response: All data in both part I and II in this study were normally distributed and authors added this information in the context at line 176 and 256.

Line 169 Please list the breed in alphabetical order and later on, too.

Response: The breed was arranged alphabetically at the reviewer's suggestion at line 175-176. 

Table 1 and Figure 2. Please write "Shih Tzu"

Response: All information was revised following the reviewer's suggestion.

Table 2 Should it be p<0.05? Please confirm.

Response: Authors revised it to be p<0.05 at line 230.

Table 3 A range is not needed for normally distrubuted values.

Response: Author removed ranges from table 3. 

Figure 3. Please add the meaning of the asterik.

Response: Author added the asterisk at the line 272  to indicate the meaning of this sign. 

Line 249: Please change heart to cardiac.

Response: Authors revised from “heart” to “cardiac” at line 284. 

Line 291 Using defined measurement points, RLAD and VLAS are objective methods like VHS.

Response: Authors revised the measurement point of these two techniques that are objective methods at line 334.

Line 293 Please explain the abbreviation LA. 

Response: Authors add the full name of LA at line 336.

Line 295 VLAS reference values for Pugs, Maltese, and Cavalier King Charles Spaniel has been published.

Response: Authors add the information of these dog breeds at line 337-339.

Line 302: Where the males Corgis larger?

Response: Author revised the context from “where” to be “in which” at line 346.

Line 312-313: Mongrels are not define and therefore not suitbale for a comparision . Please use as reference for the Laprador Retriever  Lamb et al. [2001].

Response: Authors removed Mongrel from this context at line 356-357.

Line 323-328 I think this sentence can be deleted, because it provides no necceassy information for the discussion (and LV should be explained).

Response: Author removed these sentences following reviewer’s suggestion. 

Limations

As the authors mentioned in the discussion that study which larger groups should be conducted, the population size should be mentioned as limitation. 

A second limitation is that there was no comparison between the breeds for T4L as mentioned in Line 258.

Response: Authors added the both limitations following reviewer’s suggestion at line 372-380.

A measurement example of VHS can be included.

Response: Since the VHS is a well-known cardiac measurement method and due to an amount of figure and table in this study, authors decided not to add more images but put the reference of the original method on the context. 

For example following citations can be included:

https://doi.org/10.1371/journal.pone.0274085

https://doi.org/10.1016/j.jvc.2020.06.001

https://doi.org/10.1016/j.jvc.2022.08.004

https://doi.org/10.1111/vru.13027

 https://doi.org/10.1136/vr.148.23.707

Response: Thank you very much for valuable information, authors added these references in the context and reference list.

Reviewer 2 Report

In this study, the authors are considering that the T4 dimensions, as well as the intervertebral disc space (T4-T5 IVS) may affect the VHS value, making it difficult to compare the VHS between different breeds of dogs, especially Corgi, Shih Tzus, Schnauzers, Beagles and Dachshunds. This seems to me an innovative approach, however, some points must be considered:

- The number of dogs included in part I (20 dogs of each breed) should be considered a limitation of the study.

- In addition to the T4 dimensions, VHS values for all dog breeds would be important for the results of this research. The authors mentioned that VHS was not evaluated in all dogs in part I of the study and even placed this as a limitation (linhas 335-336). However, they did not explain why. Were the radiographs not suitable for determining the VHS?

- Were the radiographic measurements taken by a single observer?

- Did the authors assess interobserver variability?

- Statistical differences described in the text (lines 171-175, age and BW) were not identified in Table 1. Why?

- In the discussion (lines 258-260), the authors stated that: "In this study, the actual T4L could not be compared among dog breeds because of the variability in the body size that may directly be correlated with the dimensions of the vertebral column." However, in the results (lines 182-185), the authors mentioned the lowest and highest values for T4L and T4H between the different dog breeds, which is still a comparison. Thus, the conclusion (lines 258-260) contradicts the results (lines 182-185).

- Table 3: the letter "a" was not indicated in the legend.

- Figures 2 and 3: considering that the mean values of T4L/H and VHS were already presented in Tables 2 and 3, respectively, the authors could modify the graphs from the inclusion of the individual values of T4L/H and VHS for each one of the dog breeds. 

- The authors cited that the hypothesis of the study was "that Corgi-specific VHSs may be similar to those of Dachshunds because of their similar body conformation and similar thoracic vertebral characteristics" (lines 81-82). Has this hypothesis been confirmed? The authors did not mention anything about this hypothesis in the discussion and conclusion of the article.

Author Response

Reviewer2

In this study, the authors are considering that the T4 dimensions, as well as the intervertebral disc space (T4-T5 IVS) may affect the VHS value, making it difficult to compare the VHS between different breeds of dogs, especially Corgi, Shih Tzus, Schnauzers, Beagles and Dachshunds. This seems to me an innovative approach, however, some points must be considered:

Response: Thank you very much for all comments and suggestions. Authors have revised and responded to all comments with the following details.

- The number of dogs included in part I (20 dogs of each breed) should be considered a limitation of the study.

Response: Authors added the both limitations following reviewer’s suggestion at line 372-380. 

- In addition to the T4 dimensions, VHS values for all dog breeds would be important for the results of this research. The authors mentioned that VHS was not evaluated in all dogs in part I of the study and even placed this as a limitation (linhas 335-336). However, they did not explain why. Were the radiographs not suitable for determining the VHS?

Response: Authors added the reason why this study did not perform the self-VHS of the dog in part 1 at line 373-376. 

- Were the radiographic measurements taken by a single observer?

Response: Yes. Authors added the information of an observer at line 104-105 and 148. 

- Did the authors assess interobserver variability?

Response: No, this study did not evaluate the interobserver variability.

- Statistical differences described in the text (lines 171-175, age and BW) were not identified in Table 1. Why?

Response: Author added the information of the significance of age and BW in table 1.

- In the discussion (lines 258-260), the authors stated that: "In this study, the actual T4L could not be compared among dog breeds because of the variability in the body size that may directly be correlated with the dimensions of the vertebral column." However, in the results (lines 182-185), the authors mentioned the lowest and highest values for T4L and T4H between the different dog breeds, which is still a comparison. Thus, the conclusion (lines 258-260) contradicts the results (lines 182-185).

Response: To make the sentence to be more understandable, authors revised the information in the discussion part to be . The actual T4L is suggested not to be compared among dog breeds at line 301.

- Table 3: the letter "a" was not indicated in the legend.

Response: Authors inserted the meaning of “a” in the legend.

- Figures 2 and 3: considering that the mean values of T4L/H and VHS were already presented in Tables 2 and 3, respectively, the authors could modify the graphs from the inclusion of the individual values of T4L/H and VHS for each one of the dog breeds. 

Response: In part I, only the T4 characteristic was evaluated, not the VHS for each dog breed. Therefore, the authors decided not to combine the graph since there was no VHS information of other dog breeds in this study.

- The authors cited that the hypothesis of the study was "that Corgi-specific VHSs may be similar to those of Dachshunds because of their similar body conformation and similar thoracic vertebral characteristics" (lines 81-82). Has this hypothesis been confirmed? The authors did not mention anything about this hypothesis in the discussion and conclusion of the article.

Response: To make the information be more understandable, authors revised the hypothesis to be “The hypothesis of this study was that Corgi-specific VHSs may be similar to those of previous reported of Dachshund’s VHS because of their similar body conformation and similar thoracic vertebral characteristics” at line 80-82 and the information that discuss the result with Dachshund was indicated at line 324-332.

Reviewer 3 Report

The article deals with a very interesting and clinically important topic. Both the results and the discussion are well written and understandable. However, the introduction needs improvement. A chest X-ray is not a test that detects heart disease, because we will not see, for example, valvular regurgitation on an X-ray. With this test, we can detect an enlarged heart silhouette as well as congestive left-sided heart failure and assess its severity.

It is a test with which we can detect a significant heart disease (with enlargement of the heart chambers). Due to the greater availability of X-ray and lower costs, this examination is very often used in cardiology.

The Corgi dog breed, like other small and medium-sized dogs, is predisposed to chronic mitral valve disease and not, as the authors wrote, to hypertrophic cardiomyopathy. CMVD affects small and medium-sized breeds dogs in old age. This also needs to be corrected and clarified in the introduction of the article.

Author Response

Reviewer3

The article deals with a very interesting and clinically important topic. Both the results and the discussion are well written and understandable. However, the introduction needs improvement. A chest X-ray is not a test that detects heart disease, because we will not see, for example, valvular regurgitation on an X-ray. With this test, we can detect an enlarged heart silhouette as well as congestive left-sided heart failure and assess its severity.

Response: Authors revised the context following reviewer’s suggestions at line 44-45.

It is a test with which we can detect a significant heart disease (with enlargement of the heart chambers). Due to the greater availability of X-ray and lower costs, this examination is very often used in cardiology.

The Corgi dog breed, like other small and medium-sized dogs, is predisposed to chronic mitral valve disease and not, as the authors wrote, to hypertrophic cardiomyopathy. CMVD affects small and medium-sized breeds dogs in old age. This also needs to be corrected and clarified in the introduction of the article.

Response: To make the context to be clearer, authors removed hypertrophic cardiomyopathy from line 74. 

Reviewer 4 Report

The article is well written.

The most interesting part is the prospective study carried out on the Corgis. The Corgi, being a breed, with a very unique chest conformation, shows significant differences in the VHS compared to other small breeds, but with different conformation. This data is useful for the clinician to evaluate the chest radiographs of Corgis and underlines that each breed, especially those that differ so much from mesomorphic dogs, must be evaluated with specific breed parameters.

Author Response

Reviewer4

The article is well written.

The most interesting part is the prospective study carried out on the Corgis. The Corgi, being a breed, with a very unique chest conformation, shows significant differences in the VHS compared to other small breeds, but with different conformation. This data is useful for the clinician to evaluate the chest radiographs of Corgis and underlines that each breed, especially those that differ so much from mesomorphic dogs, must be evaluated with specific breed parameters.

Response: Thank you very much for your comments. 
